# SGC-CAMKK2-1: A Chemical Probe for CAMKK2

**DOI:** 10.3390/cells12020287

**Published:** 2023-01-11

**Authors:** Carrow Wells, Yi Liang, Thomas L. Pulliam, Chenchu Lin, Dominik Awad, Benjamin Eduful, Sean O’Byrne, Mohammad Anwar Hossain, Carolina Moura Costa Catta-Preta, Priscila Zonzini Ramos, Opher Gileadi, Carina Gileadi, Rafael M. Couñago, Brittany Stork, Christopher G. Langendorf, Kevin Nay, Jonathan S. Oakhill, Debarati Mukherjee, Luigi Racioppi, Anthony R. Means, Brian York, Donald P. McDonnell, John W. Scott, Daniel E. Frigo, David H. Drewry

**Affiliations:** 1Structural Genomics Consortium, UNC Eshelman School of Pharmacy, University of North Carolina at Chapel Hill, Chapel Hill, NC 27599, USA; 2Department of Cancer Systems Imaging, The University of Texas MD Anderson Cancer Center, Houston, TX 77054, USA; 3Centro de Química Medicinal (CQMED), Centro de Biologia Molecular e Engenharia Genética (CBMEG), Universidade Estadual de Campinas (UNICAMP), Campinas 13083-886, Brazil; 4Department of Molecular and Cellular Biology, Baylor College of Medicine, Houston, TX 77030, USA; 5St Vincent’s Institute of Medical Research, Fitzroy, VIC 3065, Australia; 6Drug Discovery Biology, Monash Institute of Pharmaceutical Sciences, Parkville, VIC 3052, Australia; 7Department of Pharmacology and Cancer Biology, Duke University School of Medicine, Durham, NC 27705, USA; 8Department of Medicine, Division of Hematological Malignancies and Cellular Therapy, Duke University School of Medicine, Durham, NC 27710, USA; 9Department of Molecular Medicine and Medical Biotechnology, University of Naples Federico II, 80131 Naples, Italy; 10The Florey Institute of Neuroscience and Mental Health, Parkville, VIC 3052, Australia; 11Department of Genitourinary Medical Oncology, The University of Texas MD Anderson Cancer Center, Houston, TX 77030, USA; 12Center for Nuclear Receptors and Cell Signaling, University of Houston, Houston, TX 77204, USA; 13Department of Biology and Biochemistry, University of Houston, Houston, TX 77204, USA; 14Lineberger Comprehensive Cancer Center, Department of Medicine, School of Medicine, University of North Carolina at Chapel Hill, Chapel Hill, NC 27599, USA

**Keywords:** CAMKK2, kinase, chemical probe, NanoBRET, kinase selectivity

## Abstract

The serine/threonine protein kinase calcium/calmodulin-dependent protein kinase kinase 2 (CAMKK2) plays critical roles in a range of biological processes. Despite its importance, only a handful of inhibitors of CAMKK2 have been disclosed. Having a selective small molecule tool to interrogate this kinase will help demonstrate that CAMKK2 inhibition can be therapeutically beneficial. Herein, we disclose SGC-CAMKK2-1, a selective chemical probe that targets CAMKK2.

## 1. Introduction

Kinases have been successfully targeted by medicinal chemistry programs to develop therapies for diseases, mostly directed towards oncology indications. Although there are over 60 approved small molecule inhibitors targeting kinases (https://brimr.org/protein-kinase-inhibitors/; accessed 9 January 2023), many of the >500 protein kinases have garnered little attention and have no published selective or potent inhibitors. CAMKK2 (Calcium/Calmodulin-Dependent Protein Kinase Kinase 2) is a kinase that is important in physiological process such as energy balance [1] and many pathological processes (Table 1). For example, CAMKK2 activation has been linked to several types of cancers, metabolic diseases, and regulation of the immune response (Table 1). Despite having multiple links to disease, and thus a potential therapeutic target, there is only one tool molecule that is commercially available and in widespread use—STO-609 (Figure 1).

The compound STO-609 has demonstrated CAMKK2 inhibition in both cellular and in vivo settings [7,31,44]. However, it has also been shown to inhibit additional kinases such as CK2, AMPK, MNK1, PIM2, PIM3, DYRK2, DYRK3, and ERK8, which at high treatment concentrations could lead to confounding results. Recently, larger screening efforts have been undertaken to identify small molecules defined as inhibitors for other kinases that can inhibit CAMKK2 [45]. These studies were performed using a thermal shift (differential scanning fluorimetry, DSF) and enzyme inhibition assays. From these efforts, 20 compounds stabilized the protein more than 10 °C, and 10 compounds inhibited CAMKK2 in the enzyme assay at concentrations <100 nM. These compounds, while potent, inhibit additional kinases making any phenotype observed using these compounds difficult to attribute directly to CAMKK2 inhibition. To address this limitation, we sought to develop a potent and selective CAMKK2 probe molecule that could be utilized to investigate the biological implications of CAMKK2 inhibition more thoroughly. A probe molecule is a potent and selective small-molecule inhibitor that has a well-characterized mechanism of action allowing the user to attribute a biological response directly to a target [46]. In addition to the probe molecule, a structurally related but inactive compound should ideally be available to be profiled alongside the probe to give additional confidence in the biological activity of the probe.

A few other promising CAMKK2 inhibitors have been disclosed in the literature, such as a recent publication where scientists from GSK describe three related CAMKK2 inhibitor scaffolds that were optimized to afford a blood–brain penetrant inhibitor, compound **1** (**4t**) (Figure 1) [47]. These molecules, despite their single digit nanomolar CAMKK2 potency, were shown to have activity beyond CAMKK2 in the small panel they were evaluated against, making them good starting points for further optimization, but suboptimal for directly interrogating CAMKK2′s roles in biology. To obtain a molecule whose biological results could be more directly attributed to CAMKK2 inhibition, we started a medicinal chemistry campaign to identify a probe using the very potent compound **2** (GSK650394) as a chemical starting point.

## 2. Materials and Methods

### 2.1. Synthesis

General chemistry information: All reagents and solvents, unless specifically stated, were used as obtained from their commercial sources without further purification. Solvents were degassed with nitrogen for cross-coupling reactions. Air and moisture sensitive reactions were performed under an inert atmosphere using nitrogen in a previously oven-dried reaction flask, and addition of reagents were done using a syringe. All microwave (µW) reactions were carried out in a Biotage Initiator EXP US 400W microwave synthesizer. Thin layer chromatography (TLC) analyses were performed using 200 μm pre-coated sorbtech fluorescent TLC plates and spots were visualized using UV light. High resolution mass spectrometry samples were analyzed with a ThermoFisher Q Exactive HF-X (ThermoFisher, Bremen, Germany) mass spectrometer coupled with a Waters Acquity H-class liquid chromatograph system. All HRMS were obtained via electrospray ionization (ESI). Column chromatography was undertaken with a Biotage Isolera One or Prime instrument. Nuclear magnetic resonance (NMR) spectrometry was run on a Varian Inova 400 MHz or Bruker Avance III 700 MHz spectrometer equipped with a TCI H-C/N-D 5 mm cryoprobe and data were processed using the MestReNova processor. Chemical shifts are reported in ppm with residual solvent peaks referenced as internal standard.

### 2.2. Eurofins DiscoverX Broad Kinome Profiling (KINOMEscan™)

Compounds were screened at Eurofins DiscoverX (Freemont, CA) at a single concentration of 1 μM using binding assays as described previously [48,49]. Briefly, extracts containing DNA-tagged kinases are incubated with immobilized kinase inhibitors and the test compound (at 1 μM). Test compounds that displace the immobilized broad-spectrum inhibitors are detected by quantitative PCR. The results are expressed as the percentage of kinase bound to the bead compared to DMSO control (%control). Compounds with high affinity have %control of <10. A measure of selectivity is the S_10_ (1 μM) which provides a measure of kinases that demonstrate PoC < 10 at a particular concentration, in this case 1 μM. The formula to calculate S10 depicted in Equation (1).
(1)$$S_{10}\left(1\;\mu M\right)=\frac{number\;of\;wild\;type\;kinase\;with\;\%\;control\;\leq 10}{number\;of\;wild\;type\;kinase\;tested}$$

### 2.3. CAMKK2 Enzyme Assay

CAMKK2 activity was determined by measuring the transfer of radiolabeled phosphate from [γ-^32^P]-ATP to a synthetic peptide substrate (CaMKKtide) as previously described [50]. Briefly, purified recombinant CAMKK2 (100 pM) was incubated in assay buffer (50 mM HEPES [pH 7.4], 1 mM DTT, 0.02% [*v*/*v*] Brij-35) containing 200 μM CaMKKtide (Genscript), 100 μM CaCl_2_, 1 μM CaM (Sigma-Aldrich, Castle Hill, NSW, Australia), 200 μM [γ-^32^P]-ATP (Perkin Elmer, Boston, MA, USA), 5 mM MgCl_2_ (Sigma-Aldrich, Castle Hill, NSW, Australia), and various concentrations of inhibitors (0–1 μM) in a standard 30 μL assay for 10 min at 30 °C. Reactions were terminated by spotting 15 μL onto P81 phosphocellulose paper (GE Lifesciences, Paramatta, NSW, Australia) and washing extensively in 1% phosphoric acid (Sigma-Aldrich, Castle Hill, NSW, Australia). Radioactivity was quantified by liquid scintillation counting.

### 2.4. NanoBRET Cellular Target Engagement

CAMKK2 NanoBRET assay: To quantify the cellular activity of these inhibitors we developed a CAMKK2 NanoBRET target engagement assay [51,52]. Briefly this assay utilizes a nanoluciferase (NL) fused to the kinase domain of CAMKK2. This NL kinase fusion is then transiently transfected into HEK293 cells and after 24 h, tracer is added to the cells. When the tracer and the NL-CAMKK2 fusion come into proximity they create a BRET signal that can be competed in a dose-dependent manner by the addition of cell-penetrant CAMKK2 inhibitors.

### 2.5. In Silico Docking of SGC-CAMKK2-1

SGC-CAMKK2-1 was docked onto the available crystal structure of CAMKK2 bound to a similar furopyridine compound (**13g** from PMID: 34264658/PDB ID 5UY6) [52]. Co-crystal structure coordinates were downloaded from the Protein DataBank and prepared for docking using the protein preparation workflow in Maestro (Schrödinger. LLC, New York, NY, USA; version 13.0.137) using default settings [53]. Briefly, missing side chains for residues Glu277, Glu361, Arg363, and Glu439 were filled in (no atoms in these residues were <10 Å from the ligand); missing residues (214-221) were not filled in and protein termini were not capped; potential hydrogen bond assignments were optimized using PROPKA [54]; energy minimization was performed (heavy atoms were restrained with a harmonic potential of 25 kcal mol^−1^ Å^−2^; hydrogens were not restrained) using the OPLS4 force field; water molecules >5 Å from ligand atoms were deleted. The receptor grid was also generated in Maestro using default options. Briefly, van der Waals radius scaling factor was set to 1.0 and partial charge cutoff to 0.25; the docking box was limited to a 10 Å cube defined around the centroid of the ligand; grid-based constraints used were: hydrogen bond (main chain NH of Val270 and side chain NH of Lys194) and positional (defined as a 5 Å sphere around the ligand furopyridine moiety). The ligand (SGC-CAMKK2-1) was prepared for docking using LigPrep within Maestro using default settings. Briefly, the OPLS4 force field was used and possible protonation states were generated at pH 7.0 ± 2.0 using Epik. Docking of SGC-CAMKK2-1 onto CAMKK2 coordinates was performed using Glide within Maestro [55] using default settings. Briefly, a van der Waals radii scaling factor of 0.8 and a partial charge cutoff of 0.15 was used; all three grid-based constraints (2x hydrogen bonds and 1x positional) as described above were enforced; docking was performed at extra precision (XP mode) with flexible ligand sampling (including nitrogen inversion and ring conformations); and Epik state penalties were added to the final docking score (−13.7565).

### 2.6. Western Blot Analysis

#### 2.6.1. Prostate Cancer Cellular Screening

C4-2 cells were plated in 6-well plates in IMEM medium containing 0.5% FBS. After 72 h, the cells were then treated with the compounds for 24 h before the media was aspirated and cells were washed twice in ice-cold PBS. Cells were lysed using RIPA buffer containing phosphatase and protease inhibitor cocktail while rotating for 30 min at 4 °C. In each lane, 30 μg/well of protein lysate was loaded into a 10% SDS-PAGE gel and run for 1 h and 30 min. Gels were then transferred overnight in a TRIS-glycine/methanol transfer buffer onto a PVDF membrane at 4 °C. Membranes were blocked, incubated with primary overnight at 4 °C, washed, incubated with secondary at room temperature for 1 h, washed, and then developed on an Azure Biosystems C-600 imager. Densitometry was performed using ImageJ. Total-AMPK was normalized to GAPDH and then p-AMPK was normalized to normalized total-AMPK. Normalized p-AMPK was used to calculate IC_50_ in GraphPad Prism 9.5.0. Primary antibodies used were from Cell Signaling (Danvers, MA, USA; Phospho-AMPKα (Thr172) (40H9) Rabbit mAb: Cat#: 2535; AMPKα (D5A2) Rabbit mAb Cat#: 5831), BD Bioscience (Franklin Lakes, NJ, USA; CAMKK2 mouse mAb Cat# 610544), and Sigma (St. Louis, MO, USA; GAPDH rabbit pAb: Cat# G9545). Secondary antibody (Goat Anti-Rabbit IgG (H + L)-HRP Conjugate; Cat#:1706515) was from Bio-Rad Laboratories (Hercules, CA, USA).

#### 2.6.2. Breast Cancer Cellular Screening

MDA-MB-231 cells were treated with increasing doses of STO-609 (control), **5** and **7** (negative compound) for 24 h as indicated. The cells were then washed three times with 2 mL of ice-cold PBS and lysed with 0.15 mL of phospho-RIPA lysis buffer (Tris-Cl pH 7.5, 50 mM; NaCl, 150 mM; NP-40, 1%; sodium deoxycholate, 0.5%; SDS, 0.05%; EDTA, 5 mM; sodium fluoride, 50 mM; sodium pyrophosphate, 15 mM; ß-glycerophosphate, 10 mM; sodium orthovanadate, 1 mM) with protease inhibitor cocktail (Millipore-Sigma, P-8340). Equal amounts of protein per sample/lane were denatured and resolved by SDS-PAGE. Proteins were transferred to Odyssey Nitrocellulose Membranes (LI-COR Biosciences, cat no: 926-31092), and quantitative immunoblotting was performed using the Odyssey infrared immunoblotting detection system (LI-COR Biosciences, Lincoln, NE, USA). Images were captured using the LI-COR Odyssey Classic scanner and processed using LI-COR Image Studio software, version 4.0. Quantification of the bands was done using Image Studio software tools whereby each individual immunoblot band was highlighted to obtain the signal intensity. The signal intensity of the pAMPK band was then divided by that of total AMPK and the data were normalized to the vehicle-only controls for each experiment. Primary antibodies used were anti-phospho AMPKα (Thr172) (Cell Signaling, cat no: 2535, dilution 1:1000); anti-AMPKα (Cell Signaling, cat.no: 2532, dil 1:500), and anti β-actin (Cat no: 3700, Cell signaling, dil 1:10,000). Secondary antibodies used were CF680 goat anti-mouse IgG (Biotium, cat no: 20065; dilution 1:15000) and CF770 goat anti-rabbit IgG (Biotium, cat no: 20078; dilution 1:15000). All antibodies were used according to the manufacturer’s instructions.

### 2.7. Pharmacokinetic Analysis

#### 2.7.1. Animals

CD-1 mice with weights 23–25 g were purchased from Charles River Laboratories.

#### 2.7.2. Animal Experiments

The animal experiments were performed at Alliance Pharma (Malvern, PA, USA) in accordance with the guide for the care and use of laboratory animals. Alliance Pharma is accredited by the Association for Assessment and Accreditation of Laboratory Animal Care (AAALAC).

#### 2.7.3. Pharmacokinetic (PK) Study of Compound **5** and STO-609 in Mice

The intraperitoneal (IP) pharmacokinetics of **5** and STO-609 were evaluated by administering the compounds at a dose of 10 mg/kg via IP administration to three animals for each compound. The test compounds were dissolved in DMSO and then diluted to the appropriate concentration with the vehicle. For formulation we targeted the minimal amount of DMSO that could be used in combination with a formulation composed of 0.5% HPMC/0.2% Polysorbate 80. The final formulation vehicle for probe **5** was 5% DMSO in 0.5% HPMC/0.2% Polysorbate 80 and for STO-609 was 20% DMSO in 0.5% HPMC/0.2% Polysorbate 80. Blood was collected at t = 0.5, 1, 3, and 8 h and immediately spun for 3 min, the plasma was collected, and compound levels were quantified with HPLC/MS. The raw data are available in the Appendix A.

## 3. Results

### 3.1. CAMKK2 Probe Development Synthesis and Profiling

To address the lack of a highly selective CAMKK2 inhibitor and provide the community with an additional tool to cross-validate work done using STO-609, we initiated a medicinal chemistry campaign to identify and fully characterize a CAMKK2 chemical probe. We used compound **2** (Figure 1) from Price et al.[47] as our chemical starting point, reasoning that we could improve the selectivity of this very potent starting point (pIC_50_ = 9.2). We hypothesized that modifying the pyrrolo [2,3-b]pyridine hinge-binding scaffold, common to many kinase inhibitors, would provide the best opportunity to increase selectivity. Our strategy to identify scaffolds with improved selectivity has been reported [52]. In short, modification of the hinge binder from the pyrrolo [2,3-b]pyridine to a furo [2,3-b]pyridine core provided the boost of selectivity needed. Furopyridine cores are less prevalent amongst literature kinase inhibitors, perhaps because they provide one less interaction with the hinge region of the kinase. We opted to maintain the benzoic acid that made a key interaction with the catalytic lysine in hopes that would aid in maintaining potency. The route we employed (Figure 1, Appendix A) started with the 5-chlorofuro [2,3-b]8yridine-3-triflate (**8**). This route had been previously optimized by our group and provided a furopyridine core that could be quickly modified at both the 3- and 5-positions [56]. Compound **8** then underwent a Suzuki–Miyara reaction with **9** to preferentially install the cyclopentylbenzoic acid methyl ester at the 3-position (**10**). Having the chloro handle at the 5-position, we then performed a subsequent Suzuki–Miyara reaction with an aryl boronic acid to obtain the desired intermediates. Hydrolysis of the methyl ester to the carboxylic acid proceeded as expected to provide final compounds (**5** and **6**). The synthesis of the pinacol borane **9** was previously described [56].

A series of furo [2,3-b]pyridines were generated with modifications off the core at the 3- and 5-positions. They were profiled in the CAMKK2 enzyme assay we have described and utilized previously. Briefly, the enzyme assay measures the ability of the compound to inhibit the transfer of a radiolabeled phosphate from ATP to a CAMKK2-specific peptide. One compound in the series, compound **5** (SGC-CAMKK2-1), although less potent than azaindole GSK650394, retained an acceptable potency (IC_50_ = 30 nM). **5** was then profiled in the DiscoverX KINOME*scan*™ assay panel to provide a view of the overall kinome-wide selectivity (Appendix A). This assay platform evaluates compound affinity for a panel of more than 400 wild-type human kinases. The S_10_ (1 µM), a calculated score that is reflective of broad kinome selectivity, is 0.002 for the probe molecule. In this panel, compound **5** only demonstrated significant affinity for CAMKK1 and CAMKK2 (CAMKK2 PoC = 5.4, CAMKK1 PoC = 12) at a compound treatment of 1 µM (Figure 2). The affinity for the next best kinase, MYO3B, is significantly lower at only 45% of control.

### 3.2. Development of a Structurally Related Negative Control

We next sought to develop a structurally related but CAMKK2-inactive compound to serve as a negative control [57]. Two minor modifications to the probe (**5**), removal of the methyl group from the 5-position aryl ring and replacement of the cyclopentyl moiety with a chloro resulted in negative control compound **7** (SGC-CAMKK2-1N). These changes rendered this compound significantly less potent with CAMKK2 IC_50_ = 27 µM (Figure 3). In the DiscoverX KinomeSCAN^®^ assay panel, when screened at a concentration of 1 µM, compound **7** bound to very few kinases with S_10_ (1 µM) = 0.005. The only two kinases with significant binding for compound 7 were PIP4K2C with a PoC = 7.2 and PIM2 with PoC = 7.4. PIP4K2C is a lipid kinase that has very little catalytic activity [58,59,60], and there is no commercially available kinase activity assay, so this potential off-target activity for the negative control remains uncharacterized. We evaluated the PIM2 inhibition of negative control compound **7** at Reaction Biology Corporation and it had an IC_50_ = 1.9 µM.

The synthetic route to access the negative control (Figure 2, Appendix A) commenced with a Suzuki–Miyaura reaction between commercially available material 5-bromofuro [2,3-*b*]pyridine **13** and phenylboronic acid to yield the aryl-substituted derivative **14**, which was treated with bromine to afford **15** [61]. Reaction of **15** with 4-borono-2-chlorobenzoic acid afforded the carboxylic acid **7**.

### 3.3. Molecular Basis for Ligand Binding

We have previously published the crystal structure of compound **6**, a close analog of the chemical probe SGC-CAMKK2-1, bound to the CAMKK2 kinase domain (KD) (amino acids 161-449) [52]. The overall structure of the complex is depicted in Figure 4. Figure 4A highlights the detailed interactions of the close analog compound **6**, and Figure 4B shows the predicted binding mode of the probe SGC-CAMKK2-1. The two compounds (the probe and compound **6**) differ only by the presence of a methyl group in the meta position of the aryl group attached to the pyridine ring in the probe SGC-CAMKK2-1. The co-crystal structure of compound **6** bound to CAMKK2 KD revealed the ligand carboxylate moiety participates in an extensive hydrogen bond network with residues from catalytically important regions of the kinase domain, such as Asp330 in the DFG motif, Glu263 in α-helix C, and Lys194. Some of these interactions are mediated by water molecules. The ligand makes an additional hydrogen bond to the main chain N atom of Val270 in the kinase domain hinge region. Given the structural similarities between the two compounds, we expect a near identical binding mode for the probe. Indeed, in silico docking of SGC-CAMKK2-1 onto the structure of CAMKK2 KD bound to **6** indicated the kinase binds to both ligands in a similar manner. In the predicted binding pose, the additional methyl group in SGC-CAMKK2-1 is sandwiched between the side chain of Pro274 and the main chain of Gly172. This latter residue is part of the P-Loop region in the kinase domain (for clarity the P-Loop is not shown in Figure 4).

Our binding and structural data suggest that the selectivity of **6** is a function of its single hinge contact and its ability to mediate both hydrophobic and polar interactions with CAMKK2′s ATP-binding site. Most ATP-competitive inhibitors engage the kinase hinge region via two or more hydrogen bonds to protein main chain atoms. Reducing the number of contacts between the inhibitor and the protein has been suggested as a strategy to increase selectivity [62,63].

### 3.4. CAMKK2 NanoBRET in Cell Target Engagement

In addition to in vitro enzymatic potency and kinome selectivity, a chemical probe needs to have suitable cellular potency. To evaluate this, we developed a NanoBRET in-cell target engagement assay. For this assay, the CAMKK2 kinase domain fused with a Nanoluciferase (NLuc) tag is transiently transfected into HEK293 cells. After 24 h, a heterobifunctional molecule that consists of an inhibitor that binds to CAMKK2 linked to a BODIPY dye is added to the cells. When the inhibitor–dye hybrid binds to the kinase, the proximity to the NLuc fused to CAMKK2 creates a BRET signal. This signal can then be competed away in a dose-dependent fashion with free inhibitor. Using this assay, we evaluated the probe and STO-609 (Figure 5). SGC-CAMKK2-1 (**5**) was able to compete away the tracer with an in-cell target engagement IC_50_ of 270 nM. In the same assay, STO-609 had an IC_50_ > 10 µM. These data demonstrate that the probe compound can engage CAMKK2 in cells at concentrations below 1 µM.

### 3.5. On-Target Cellular Effect (Western Blot)

With a potent (enzyme assay) and selective (KINOME*scan*) compound in hand, along with evidence of in-cell target engagement (nanoBRET assay), we next looked for functional consequences of CAMKK2 inhibition in cellular contexts. To do this, we evaluated the impact of our probe, the negative control, and a positive control (STO-609) on AMPK phosphorylation in C4-2 prostate cancer cells to provide evidence of a functional on-target effect. As AMPK is a direct substrate of CAMKK2, inhibition of AMPK phosphorylation at Thr172 should be observed with the CAMKK2 chemical probe and STO-609, but not the negative control. In this experiment, IC_50_ was calculated via quantifying changes in p-AMPK(Thr172) normalized to total AMPK levels. The chemical probe indeed decreased p-AMPK levels, with an IC_50_ = 1.6 µM. As a comparison, the canonical CAMKK2 inhibitor, STO-609, also decreased p-AMPK, but was roughly 7-fold weaker with an IC_50_ = 10.7 µM. Importantly, the structurally similar negative control (**7**) did not have any measurable activity (Figure 6c) in this assay.

To confirm our cellular effects in a different model, we treated MDA-MB-231 triple-negative metastatic breast cancer cells with STO-609, probe SGC-CAMKK2-1, and negative control SGC-CAMKK2-1N. Again, SGC-CAMKK2-1 was more effective than STO-609 in blocking AMPK phosphorylation, and the negative control showed no effect at a concentration of 10 μM (Figure 7).

### 3.6. Pharmacokinetic Studies

With the probe molecule having favorable selectivity and potency in vitro and in a cellular context, we next wanted to determine its suitability for in vivo experiments. SGC-CAMKK2-1 (**5**) was progressed to a single dose mouse intraperitoneal (I.P) experiment. The mice were treated with either the probe or STO-609 at a concentration of 10 mg/kg.

Figure 8 shows the plasma concentration of both SGC-CAMKK2-1 and STO-609 over time. STO-609 has considerably higher plasma concentrations than SGC-CAMKK2-1 in this experiment. Although it may be possible to find a dosing regimen that allows for evaluation of this compound in vivo, based on these results we suggest only using SGC-CAMKK2-1 in cell-based assays. Further medicinal chemistry work will be needed to optimize for in vivo use.

## 4. Discussion

A growing body of literature indicates both increased roles for CAMKK2 in a range of biological processes and a corresponding interest in the field. For example, CAMKK2 is a promising therapeutic target in a diverse set of diseases [29,40,64,65], including multiple cancers [7,15,19,66,67,68,69]. To date, STO-609 has been the primary chemical tool used to explore the consequences of CAMKK2 inhibition in models of disease. Although it has demonstrable on-target CAMKK2 activity and has had utility in understanding CAMKK2 roles in signaling, these studies remain limited by virtue of the observed off-target activities of STO-609 [45]. Here, we describe a new chemical probe, SGC-CAMKK2-1, that can be used as an additional tool to explore CAMKK2 functions in health and disease. Advantages for SGC-CAMKK2-1 include its improved kinome selectivity and potency in cells. It has weaknesses as well, including inability to distinguish between CAMKK1 and CAMKK2 (just like STO-609), and poor in vivo PK in preliminary experiments. Further work should focus on identifying analogs with suitable PK properties while maintaining or improving in-cell potency and selectivity. In addition, it will be important to understand how well this compound (and STO-609) inhibit the phosphorylation of other CAMKK2 substrates such as CAMKI and CAMKIV.

SGC-CAMKK2-1 and SGC-CAMKK2-1N are commercially available through Sigma-Aldrich. As such, and with the availability of STO-609, we believe this tool kit of compounds will facilitate building a greater understanding of the true therapeutic potential of CAMKK2 inhibition. The availability of a new, readily available CAMKK2 chemical probe represents an important next step towards the ultimate goal of developing a clinical-grade CAMKK2 inhibitor to evaluate the hypothesis that inhibition of CAMKK2 will be beneficial in certain clinical settings.

## Data Availability

Data describing the characterization of the chemical probe and its negative control are provided in the Appendix A. Broad kinome screening data are also included in the Appendix A.

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
