# Peer review of "SGC-CAMKK2-1: A Chemical Probe for CAMKK2"

_cells, 2023, doi:10.3390/cells12020287_

Round 1

Reviewer 1 Report

This study reports the identification and initial characterization of SGC-CAMKK2-1, a selective inhibitor targeting CAMKK2. Given the role of CAMKK2 in various diseases including cancer, this target is of relevance for targeted therapy.  However, the limited availability of pharmacological agents targeting this kinase hampered the accumulation of sufficient proof of concept studies. Thus, the identification of a potent and selective inhibitor represents the basis for further validation of CAMKK2 as a valuable drug target.

The paper is well-written and conclusions are generally supported by the data.

Addressing the following minor issues might further improve readability:

1) blot in figure 7A is not very clear and does not seem to represent what seen in figure 7B (pAMPK at 0.5 of compound 5 is very bright)

2) Compounds 5 and 6 are mentioned but their chemical representation needs to be more clearly indicated.  In Fig. 2, SGC-CAMKK2-1 should be changed into 5 (SGC-CAMKK2-1) in panel A. Where is compound 6?

3) The fact that CAMKK1 is hit by just a 2x higher concentration, and that SGC-CAMKK2-1 is in the end a promiscuous inhibitor should be made more explicit. The title should be changed accordingly.

Reviewer 2 Report

In this work Wells and co-workers describe the synthesis and the activity of a novel and specific inhibitor of CAMKK2, which can be useful to explore the function of CAMKK2 in cellular assays.

The rational prompting this work is the current lack of inhibitors with acceptable selectivity profile , with most of known CAMKK2 inhibitors also targeting other kinases.

Overall this is an interesting paper and the experiments are well designed. Presentation can be improved.

Main criticisms:

1) The data demonstrating the selectivity of compound 5 are solely the ones obtained by the KINOMEscanX. Here the specificity of cpd 5 seems good however the assay does not directly measure inhibition potency. I would recommend to test specificity and potency of the molecule in enzymatic assays against a smaller panel of kinases, possibly by comparing compound 5 to the starting compound 2 (GSK650394).

2) Based on the predicted binding mode to the kinase, it is expected that Compound 5 is an ATP competitor inhibitor, however this was not shown directly shown- Experiments with ATP titration in enzymatic assay would aid. 

3) Fig 4 and 5 -  it is stated that experiments were done in triplicate however the curves seems to show only one experiment- are the point related to a single experiments or are the means? Please include standard deviation.

Few suggestions for improvement:

·      Whenever possible it would be nice to see direct comparison between Cpd 2 and Cpd 5  in the different biochemical and cellular assays.

·      The method used for the analysis and quantification of western blots signals should be added to M&M.

·      Figure legends should be improved and should describe all the information in the figures. i.e. in Fig 1 define pIC50 and whether it is reported in nM or mM- in fig 2 define the what is PoC

·      Fig 4A does not require colour and labelling is very small

Reviewer 3 Report

In the present manuscript Wells and colleagues present the synthesis and validation of a novel CAMKK2 inhibitor and its inactive parent molecule. The manuscript is well written and the description of both the synthesis process as well as the validation are very well described. Overall this manuscript adds a new tool to the toolbox available  for probing CAMKK2. 

My comments / suggestions are:

1. The authors generate a BRET sensor able to detect the activity of CAMKK2 in cells. The data shown would be more convincing if:

- The authors generate a BRET probe based on a mutant version of CAMKK2 (un able to bing to the inhibitors)

- The authors should show that the inactive molecule SGC-CAMKK2-1N does not affect the BRET signal.

2. In the in vivo experiments the authors show that the SGC-CAMKK2-1 PK is much lower than that of the "canonical" CAMKK2 inhibitor STO-609. However, according to the method description the two compounds were injected in vehicles that contained significantly different levels of DMSO, 5% for SGC-CAMKK2-1 and 20% for STO-609. Is it possible that their data could be affected by this discrepancy?

Round 2

Reviewer 3 Report

The authors responded to my concerns and where they did not provided an acceptable rationale of their methodological choices.